# Novel chemical inhibitor against SOD1 misfolding and aggregation protects neuron-loss and ameliorates disease symptoms in ALS mouse model

Tae-Gyun Woo [1,2], Min-Ho Yoon[1], So-mi Kang[1], Soyoung Park[1], Jung-Hyun Cho[1], Young Jun Hwang[1], Jinsook Ahn[3], Hyewon Jang[4], Yun-Jeong Shin[4], Eui-Man Jung [1], Nam-Chul Ha [3], Bae-Hoon Kim[1,2], Yonghoon Kwon [4,5✉] & Bum-Joon Park [1,2,5✉]

Amyotrophic Lateral Sclerosis (ALS) is a fatal neurodegenerative disease characterized by selective death of motor neurons. Mutations in Cu, Zn-superoxide dismutase (SOD1) causing the gain of its toxic property are the major culprit of familial ALS (fALS). The abnormal SOD1 aggregation in the motor neurons has been suggested as the major pathological hallmark of ALS patients. However, the development of pharmacological interventions against SOD1 still needs further investigation. In this study, using ELISA-based chemical screening with wild and mutant SOD1 proteins, we screened a new small molecule, PRG-A01, which could block the misfolding/aggregation of SOD1 or TDP-43. The drug rescued the cell death induced by mutant SOD1 in human neuroblastoma cell line. Administration of PRG-A01 into the ALS model mouse resulted in significant improvement of muscle strength, motor neuron viability and mobility with extended lifespan. These results suggest that SOD1 misfolding/aggregation is a potent therapeutic target for SOD1 related ALS.

[1] Department of Molecular Biology, College of Natural Science, Pusan National University, Busan, Republic of Korea. [2] Rare Disease R&D Center, PRG S&T Co., Ltd, Busan, Republic of Korea. [3] Department of Food Science, College of Agricultural and Life Sciences, Seoul National University, Seoul, Republic of Korea. [4] Department of Agricultural Biotechnology, Seoul National University, Seoul, Republic of Korea. [5] These authors jointly supervised this work: Yonghoon Kwon and Bum-Joon Park. ✉email: y_kwon@snu.ac.kr; bjark1219@pusan.ac.kr

Amyotrophic lateral sclerosis (ALS) is a lethal neurodegenerative disease[1–3], in which motor neuron is selectively eliminated by cell death in spinal cord[4–6]. Although many studies suggesting the involvement of several altered signaling pathways, such as glutamate excitotoxicity, mitochondrial dysfunction, oxidative stress and neuroinflammation, the pathogenesis of ALS is still unclear. About 5–10% of patients show inherited patterns (familial ALS; fALS[7,8]), whereas the remaining 80- 90 % are grouped in sporadic ALS (sALS[9–11]). Several genes has been suggested from ALS-related loci in fALS[9,10]. Superoxide dismutase 1 (SOD1) gene has been firstly identified in fALS[7,8]. However, any genetic risk factors or family history in sALS has not been suggested clearly until now, although ALS-related genes (TAR DNA binding protein; TDP-43, Fused in Sarcoma; FUS and Chromosome 9 open reading frame 72; C9orf72) have been linked to sALS[12–16]. Interestingly, inclusion of wild-type (WT)-SOD1 are detected in sALS patients and mouse models with WT-SOD1 overexpression[17–22]. The pathological modification of WT-SOD1 by mutant SOD1 or sALS-related proteins like TDP43 and FUS has been reported[11,14,23,24].

One of the important features of ALS is progressive disease[25,26], which means that neural cell death is propagated more over time through neural connection. Concerning this feature, the prion-like propagation has been suggested for the progression of ALS; misfolded protein converts normal protein into abnormal protein[2,11,15,17,27]. Alzheimer's disease (AD) and Parkinson's disease (PD) are showing similar phenotype[27–31]. Indeed, mutated Amyloid beta (Aβ) can convert normal Aβ into abnormal Aβ[32]. In recent, it has been reported that mutant or misfolded SOD1 also can be secreted and cause the disease progression[27,33]. Thus, the blocking of SOD1 propagation or interaction between WT-SOD1 and mutated/misfolded SOD1 would be a plausible target for ALS disease.

In this study, we showed that the aggregation and misfolding of WT-SOD1 could be induced by mutant (MT)-SOD-1, TDP-43 or cell stress inducers. We screened the aggregation inhibitors, which can maintain the dimer formation but inhibit the oligomerization of SOD1. The selected chemical, PRG-A01, worked as SOD1 inhibitor against misfolding/aggregation, which was able to inhibit the neuronal cell death caused by MT-SOD1 overexpression. Furthermore, we confirmed the therapeutic effect of PRG-A01 in vivo by addressing that it ameliorated motor neuron regression, movement disorder, and prolonged life span in SOD1$^{G93A-Tg}$ ALS mouse model.

## Results

**Misfolding or aggregation of WT-SOD1 could be induced by its overexpression, MT-SOD1 expression and stress inducers.** To test our hypothesis that MT-SOD1 overexpression may induce misfolding and aggregation of WT-SOD1, we co-transfected the vectors encoding non-tagged-WT/ MT-SOD1s into human neuroblastoma SK-N-SH cells expressing GFP-WT-SOD1. Interestingly, all mutant types and WT-SOD1 could induce the inclusions of GFP-WT-SOD1, which was distributed in cytosol (Fig. 1a and S1a). This phenomenon was repeated in human embryonic kidney HEK293 cells (Fig. S2b, c). To validate the aggregation of SOD1 by MT-SOD1s, we prepared non-tagged WT-SOD1 and GST-MT-SOD1 proteins. WT-SOD1 proteins were incubated with GST mutant proteins dose dependently in non-denaturing condition and separated in SDS-PAGE without boiling (see figure legend for detail). Even though MT-SOD1 had a self-oligomerizing or aggregating property (Fig. 1b; upper panel, each second lane), it showed that MT-SOD1 proteins could oligomerize its wild type protein directly in vitro (Fig. 1b). Next, we performed co-transfection of the vectors encoding non-tagged-

WT/MT-SOD1s in SK-N-SH cells expressing GFP-WT-SOD1. It showed that the overexpression of non-tagged-WT or MT-SOD1s turned soluble GFP-WT-SOD1 into insoluble aggregation, but not with control empty vector (Fig. 1c). In addition, we performed the cross-linking experiment using glutaraldehyde for detecting the oligomeric status of SOD1 in SK-N-SH cells and observed the oligomerization of GFP-WT-SOD1 by the overexpression of non-tagged WT and MT-SOD1 (Fig. 1d).

Next, we asked whether the aggregation of WT-SOD1 could be formed by cellular stresses. $Ca^{2+}$ is critically important to neurons as it participates in the transmission of the depolarizing signal and contributes to synaptic activity[34,35]. Therefore, we tested the effect of thapsigargin (endoplasmic reticulum $Ca^{2+}$ depletor) and BAPTA ($Ca^{2+}$ chelator) on SOD1 aggregation. Their treatment promoted the inclusions of WT-SOD1 (white arrow) in SK-N-SH cells (Fig. 1e and S1d) and showed the prominent inclusions of GFP-WT/MT-SOD1 in HEK293 cells (Fig. S1e and f). And the cross-linking experiment showed the increased oligomerization of WT and MT-SOD1 in HEK293 cells (Fig. 1f). Furthermore, thapsigargin treatment also could increase the insoluble form in WT/MT-SOD1s (Fig. 1g and S1g, respectively). Since SOD1 is also known to use Cu/Zn ion for its enzymatic activity[36,37] and several studies have shown that in both ALS patients and mice with the disease, hypoxia signals are increased compared to healthy groups[38], we tested whether these conditions were able to induce the insoluble aggregation of WT-SOD1. As shown in Fig. 1e, g, $CoCl_2$ (hypoxia-inducer), TPEN ($Zn^{2+}$ chelator) and ATN-224 ($Cu^{2+}$ chelator) could induce the inclusions (white arrow) and insoluble form of WT-SOD1. These results confirm that SOD1 can be aggregated or converted into misfolded form by its overexpression, prion-like MT-SOD1 or cellular stresses without genetic mutation. It reminded us that misfolding/aggregation of SOD1 would give an insight to solve the complicated nature of ALS as has been suggested previously[2,39].

**The candidate molecule inhibited aggregation/misfolding of SOD1.** If one of the major reason for ALS disease is SOD1 misfolding or aggregation, its blocking can be one of the plausible strategies to develop ALS drug. Actually, neutralizing antibodies against misfolded SOD1, which blocked the spread of the prion-like mutant SOD1 between cells, showed a favorable effect in SOD1 animal model[39]. To prove our concept, we established the chemical screening system based on ELISA with our proven system and chemical library[40,41]. To find inhibitors which are selectively able to block the interaction between MT-SOD1 and WT-SOD1, recombinant WT-SOD1 protein was immobilized on the 96-well ELISA and reacted with GST tagged recombinant MT-SOD1s (A4V, G37R, G85R and G93A; Fig. S2a) in the presence or absence of the chemical library. We measured the binding inhibition effect of each chemical and primarily screened the candidates, which reduced the binding affinity between WT and MT-SOD1 (Fig. 2a; full data can be available for request). And we did secondary screening to exclude the chemicals showing very weak (less than 50 %) or very strong (more than 90%) binding inhibition (such as 001, 028, 052, 055 and 035, 041, 050, respectively; Fig. 2a). The strong in vitro binding inhibition may hamper a highly stable and functionally active SOD1 dimer[42]. Thus, ten candidates were selected by these criteria including excluded control chemical Chem-001 and −035 as shown in Fig. S2b. Using the native gel analysis and cross-linking experiment, the candidates were tested on the aggregation inhibition of MT-SOD1 (G37R and G85R) in HEK293 (Fig. S2b) and G93A-SOD1 in SK-N-SH cells (Fig. S2c), respectively and we found Chem-036 could be a candidate inhibitor for MT-SOD1

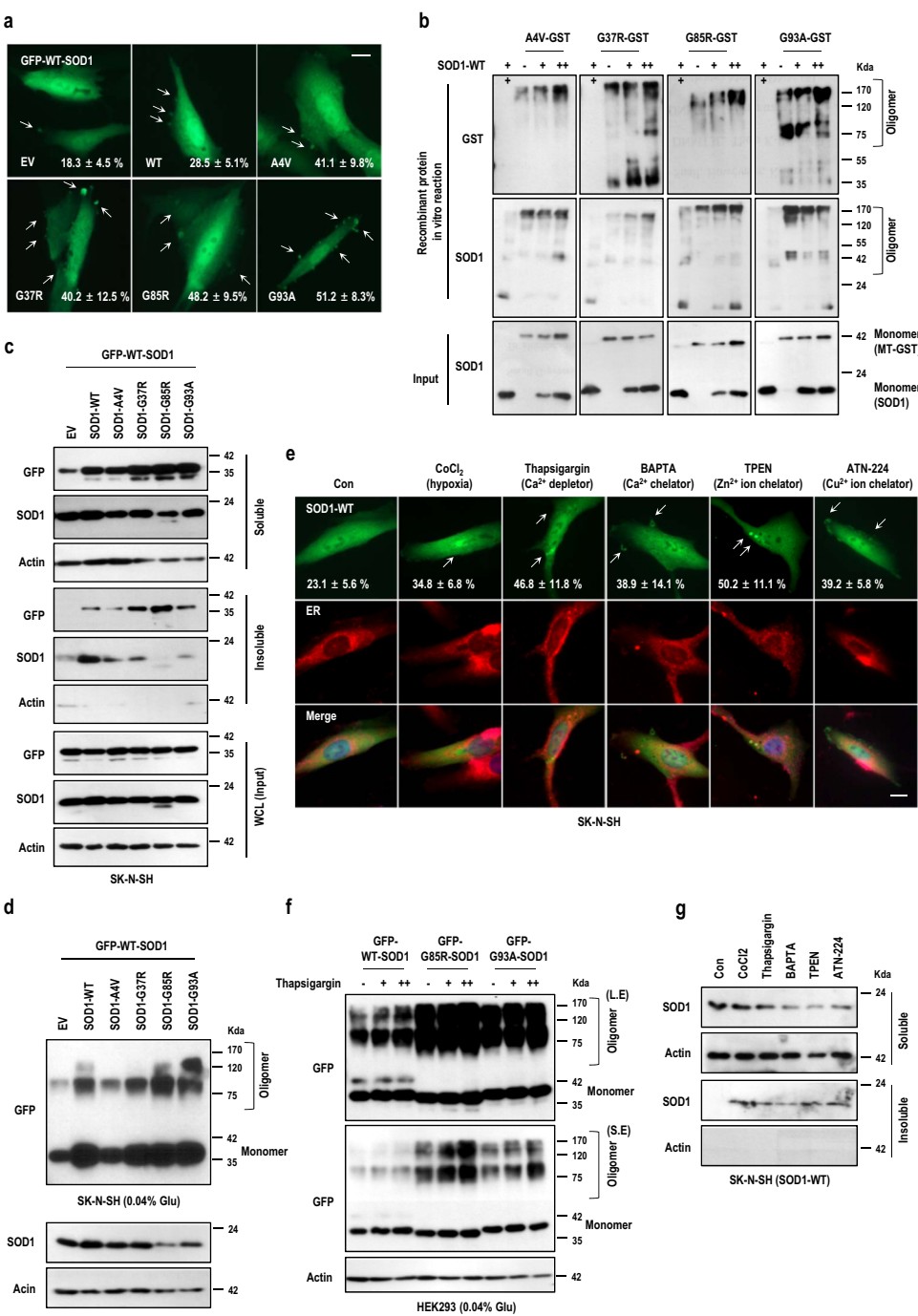

(Fig. S2d). This chemical exhibited the marginal inhibition effect on interaction between WT-SOD1s (data can be available for request). We further assessed the probable binding sites of Chem-036 through virtual screening software for computational drug discovery. As shown in Fig. S2e, Chem-036 was bound to the different sites on the surface of the dimeric structure of the native SOD1. However, all the binding energies were similar to each other, indicating that further study was needed to comprehend the function of Chem-036. We guess, Chem-036 might interact with a protein interface formed by aggregated SOD1, not with the surface of the dimeric structure. Remarkably, Chem-036 inhibited the aggregation of MT-SOD1 in a dose-dependent manner, but did not interrupt its dimer formation (Fig. 2b). And Chem-036 treatment to the cells could decrease the inclusion formation

(white arrow) of MT-SOD1, but not with other candidates (Fig. 2c, d and S2f-i). To check whether Chem-036 can reduce the insoluble and oligomer formation of WT-SOD1 by MT-SOD1 overexpression, GFP-WT-SOD1 was co-transfected into HEK293 cells expressing each MT-SOD1s. As shown in Fig. S2j and k, Chem-036 could reduce the oligomer and insoluble formation of WT-SOD1 by SOD1 overexpression, respectively. Furthermore, we confirmed that Chem-036 also inhibited the oligomer formation of WT-SOD1 by TPEN (Fig. 2e) and it reduced insoluble formation of WT and MT-SOD1 upon cellular stress by thapsigargin (Fig. S2l and m). Including thapsigargin (endoplasmic reticulum $Ca^{2+}$ depletor) and BAPTA ($Ca^{2+}$ chelator), TPEN ($Zn^{2+}$ chelator), $CoCl_2$ (hypoxia-inducer), and ATN-224 ($Cu^{2+}$ chelator) induced the inclusions in cells were significantly

**Fig. 1 Mutant SOD1 promotes WT-SOD1 aggregation. a** Mutant SOD1s induced WT-SOD1 inclusions in SK-N-SH cells. Cells expressing GFP-WT-SOD1 were co-transfected with non-tagged MT-SOD1 (SOD1-WT, A4V, G37R, G85R and G93A) for 24 h and observed under the fluorescence microscope. Cells with SOD1 inclusions (white arrows; strong intensity of SOD1) among GFP positive ones were counted by photomicrographs and the percentages were shown with standard deviation (SD). Cell counting figure was shown in Fig. S1a. $n = 3$ independent experiments; two-tailed Student's $t$-test, Scale bar; 10 μm. **b** MT-SOD1s recombinant proteins promoted oligomerization of WT-SOD1 in vitro. GST-MT-SOD1 recombinant protein (4 μg/ml) was incubated with non-tagged WT-SOD1 recombinant proteins (5, 10 μg/ml; +, ++ respectively) for 30 min at RT with PBS and the samples were mixed with non-denaturing sample buffer. Those were separated in SDS-PAGE gel without boiling. Input indicated the denaturation samples which were reacted with SDS-containing PBS. The samples were separated in SDS-PAGE gel with boiling. **c** WT and MT-SOD1s overexpression increased the insoluble form of WT-SOD1. GFP-WT-SOD1 were co-transfected with together non-tagged WT and MT-SOD1 vectors for 24 h in SK-N-SH cells. After incubation, the cells were harvested with TNN buffer (50 mM Tris-Cl, pH 7.5, 150 mM NaCl, 0.3% NP-40) and centrifuged at 14,000 rpm for 30 min then pellet (insoluble) and supernatant (soluble) were separated. Input indicated whole cell lysates (WCL) which were harvested with RIPA buffer. Actin was used as loading control. **d** WT-SOD1 was oligomerized by WT and MT-SOD1 overexpression. After transfection with indicating vectors (SOD1-A4V, G37R, G85R, G93A and GFP-WT-SOD1) in SK-N-SH cells for 24 h, the cells were harvested with TNNI (TNN buffer + 100 mM iodoacetamide) buffer and reacted with 0.04% glutaraldehyde (Glu) for 1 h. The samples were performed with SDS-PAGE. Monomer and oligomer indicated SOD1 formation. **e–g** Cellular stresses induced WT-SOD1 aggregation. **e** WT-SOD1 expression vector was transfected into SK-N-SH cells and incubated with cellular stresses {CoCl₂ (Hypoxia-inducer; 200 μM), Thapsigargin (endoplasmic reticulum Ca²⁺ depletor; 500 nM), BAPTA (Ca²⁺ chelator; 10 μM), TPEN (Zn²⁺ chelator; 2 μM), and ANT-224 (Cu²⁺ chelator; 10 μM)} for 12 h. Cells were fixed with 4% paraformaldehyde (PFA) and stained with Pan-SOD1 antibody (Green) and ER (Red) for monitoring SOD1 inclusion. Inclusion positive cells (white arrows; strong intensity of SOD1) were counted from photomicrographs and the percentages were shown with ± SD. Cell counting figure was shown in Fig. S1d. $n = 3$ independent experiments; two-tailed Student's $t$-test, Scale bar; 10 μm. **f** After transfection with indicating vectors (WT, G85R and G93A-SOD1-GFP) in HEK293 cells for 24 h and incubated with thapsigargin dose dependently (200, 500 nM). Cells were harvested with TNNI buffer and reacted with 0.04% glutaraldehyde (Glu) for 1 h. The samples were performed with SDS-PAGE. Monomer and oligomer indicated SOD1 formation. S.E: short exposure, L.E: long exposure. **g** SK-N-SH cells transfected with WT-SOD1 expression vector were treated with cellular stresses. After incubation, the cells were harvested with TNN buffer and centrifuged at 14,000 rpm for 30 min then Pellet (insoluble) and supernatant (soluble) were separated.

reduced by Chem-036 treatment (Fig. 2f, g, S3a and b). Moreover, Chem-036 did not show the cytotoxic effect in normal human fibroblasts (Fig. S2n). To address whether Chem-036 could inhibit the neuronal cell death, which was induced by MT-SOD1 overexpression, we transfected human neuroblastoma cell (SK-N-MC and SK-N-SH) with expression vector constructs for each of SOD1 mutants. As shown in Fig. S3c–f, cell death was significantly increased in both cells expressing A4V-, G37R-, G85R-, and G93A-SOD1, on the other hand cells expressing WT-SOD1 induced cell death with basal level. Conspicuously, Chem-036 treatment blocked the cell death induced by MT-SOD1 overexpression, with cell viability comparable to cells expressing WT-SOD1 (Fig. S3c–f). These results suggest that the chemical might resolve SOD1 from a toxic misfolded status form to a physiological folding form, although more intensive study should be performed for its mechanistic property. Here, we chose Chem-036 as the first hit compound for the resolving agent against SOD1 aggregation. From now, Chem-036 is re-named as PRG-A01.

**PRG-A01 inhibited the aggregation formed between SOD1 and TDP-43.** Recently, it has been reported that overexpression or deletion of TDP-43 can induce SOD1 misfolding, and cytosolic mislocalization of TDP-43 may kindle WT-SOD1 misfolding in non-SOD1 fALS and sALS[14–16]. Therefore, we checked the effect on the aggregation of WT-SOD1 induced by TDP-43 overexpression in SK-N-SH. We observed that TDP-43 overexpression alone could induce its cytosolic mislocalization and aggregation (Fig. 3a). And if it was overexpressed together with WT-SOD1, it became more disperse in cytoplasm and induced more inclusions of WT-SOD1 in SK-N-SH cells (Fig. 3a; white arrow and Fig. S4a). However, if the cells were treated with PRG-A01, the mislocalization of TDP-43 and the inclusion of WT-SOD1 and TDP-43 were remarkably disappeared (Fig. 3a–d). Ectopic TDP-43 overexpression in SK-N-SH cell made it run with high molecular complex size together with endogenous SOD1, but PRG-A01 treatment broke the complexes and reduced their association in a native PAGE gel (Fig. 3e). Cross-linking experiment using glutaraldehyde indeed showed that TDP-43

overexpression induced higher oligomeric formation of endogenous SOD1 and itself. However, PRG-A01 treatment notably diminished their oligomeric and insoluble formation of SOD1 and TDP-43 in SK-N-SH cell (Fig. 3f). And cell fractionation experiment confirmed PRG-A01 treatment reduced the insoluble formation of TDP-43, which was induced by its ectopic overexpression (Fig. S4b)

In recent, intracerebral antibodies against misfolded SOD1 have been shown to ameliorate disease phenotypes in transgenic mouse models overexpressing ALS-causing SOD1 mutations[39]. To define whether PRG-A01 could block SOD1 misfolding, we performed the dot blot analysis with cell lysates expressing WT- and MT-SOD1. Cell lysate transfected with WT or MT-SOD1 was directly transferred on the membrane by vacuum and reacted with misfolding specific SOD1 antibody (mis-SOD1 Ab). PRG-A01 could block the interaction between mis-SOD1 Ab and WT or MT-SOD1 in cell lysate (Fig. S4c and d). Figure 3g, h showed that the antibody affinity to mutant SOD1 in cell lysate was diminished by dose dependent treatments of PRG-A01. To examine whether PRG-A01 could directly protect mis-SOD1 Ab from binding the misfolded SOD1 proteins, purified WT-SOD1 or MT-SOD1-G93A or both proteins were incubated with the antibody in the condition, which induced the oligomerization or aggregation of MT-SOD1 or WT-SOD1 by MT-SOD1 as shown in Fig. 1b. mis-SOD1 Ab barely detected WT-SOD1 alone, but did well G93A mutant protein alone (Fig. 3i; each panel, each first lane). Notably, PRG-A01 could block mis-SOD1 Ab binding to the mutant SOD1 proteins and WT-SOD1 proteins with dose dependent manner (Fig. 3i; middle and lower panels, each second and third lane). It showed the candidate inhibitor, PRG-A01, might interact with a protein interface formed by aggregated SOD1, not with intact wild type SOD1. But further study is required to understand the nature of PRG-A01 on SOD1 better.

**Therapeutic effect of the PRG-A01 on the SOD1^G93A-Tg mouse model.** To address the in vivo chemical effect of PRG-A01, we injected it intraperitoneally (i.p) to SOD1^G93A-Tg model mouse, which was designed to overexpress mutant form of human superoxide dismutase 1, SOD1-G93A[20,43,44] (Fig. 4a and S5a;

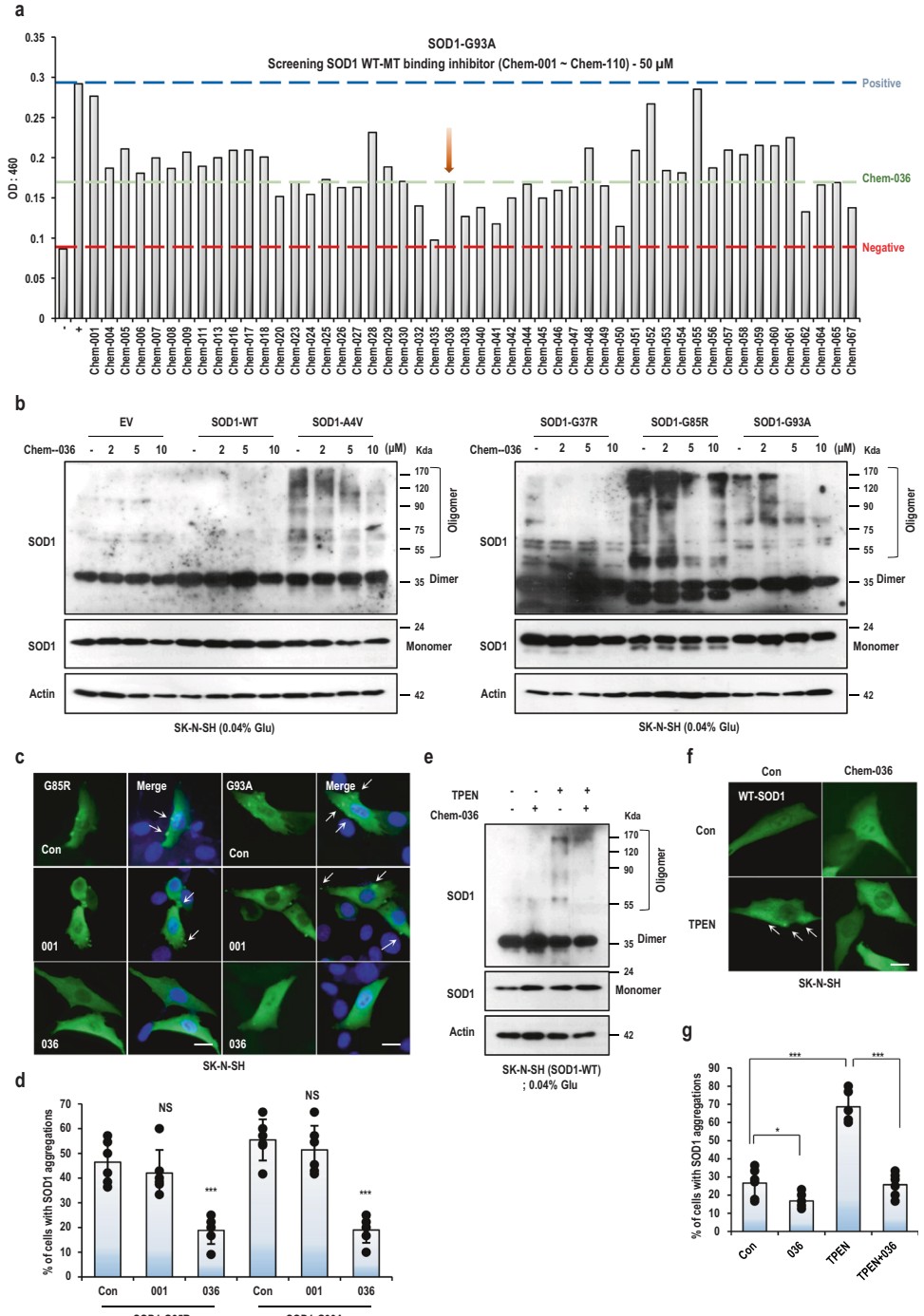

twice/week). First, the behavior of SOD1-G93A mice injected with PRG-A01 was monitored together with vehicle treated one. Until 18–19 weeks (126–133 days), PRG-A01 treated model mice still possessed the moving ability with sex-independency (Fig. S5b and Supplementary Movie 1–5). Moreover, PRG-A01 did not show toxicity about body weight loss (Fig. 4b). Vehicle-treated model mice did not stand and showed the problem in self-breathing at the terminal stage of disease (Supplementary Movie 1–5). Using open field test, which is a common measure of exploratory behavior and general activity in both mice and rats, we monitored both the quality and quantity of the activity of the model mice. As shown in Fig. 4c–e, S5c–e and S6, the activity of PRG-A01 treated model mice was remarkably maintained compared to vehicle treated mice in velocity and total distances of

movement. Especially, the movement pattern crossing the cage of drug treated mice was distinct feature from vehicle treated one, which majorly stayed on the wall side from 16 weeks (Fig. 4c, S5c and S6). Next, to test the neuromuscular function as maximal muscle strength of forelimbs, the grip strength was measured with age-matched littermates from wild type, vehicle treated model mouse and PRG-A01 treated model mouse. Vehicle treated model mice retained less than 40 % muscle strength, but PRG-A01 treated model mice about 50~60 % muscle strength compared to wild type mouse (Fig. S5f). It implicated that its treatment helped delaying the progressive muscle loss of the model mice.

Characteristic positive cytoplasmic accumulation of SOD1 with vacuolar structure (black arrowheads) in motor neurons[39,45] was observed in the ventral horn of the cervical spinal cord from

**Fig. 2 Identification of SOD1 WT-MT binding inhibitor through ELISA-based chemical screening. a** Representative graph for chemical screening. Chemicals (about 150 chemicals) were incubated with immobilized WT-SOD1 and MT-SOD1. The inhibitory effect of each chemical was determined by ELISA. Negative control (−; red line) was reacted without mutant SOD1 and positive control (+; blue line) was incubated with mutant SOD1 without chemical. The results about other chemicals and other pairings (WT-G85R, WT-A4V, WT-G37R, and WT-WT) were available if request. **b** Chem-036 suppressed SOD1 aggregation in dose dependently. After transfection with indicating vectors (WT- and MT-SOD1) in SK-N-SH cells, selected chemicals were treated for 24 h. After incubation, the cells were harvested with TNN buffer and reacted with 0.04% glutaraldehyde (Glu) for 1 h. SDS-PAGE was performed using the pan-SOD1 antibody. Monomer, dimer and oligomer indicated SOD1 formation. Chem-036 obviously suppressed the oligomerization of SOD1, but not dimer formation induced by MT-SOD1. **c-d** Chem-036 reduced SOD1 inclusions in SOD1-G85R, G93A overexpressed SK-N-SH cells. **c** Cells were transfected with SOD1-G85R, G93A vectors for 24 h. After incubation, cells were treated with Chem-001 (5 μM) and Chem-036 (5 μM) for 24 h and fixed with PFA. SOD1 expression was observed under fluorescence microscope with pan-SOD1 antibody (white arrows; strong intensity of SOD1). Chem-001 showed the marginal effect on inhibition of SOD1 inclusions. Scale bar; 10 μm. **d** Cells with SOD1 inclusions (white arrows; strong intensity of SOD1) were counted by photomicrographs and the percentages were shown with ± SD. $n = 3$ independent experiments; two-tailed Student's $t$-test, NS; Not significant, ***$P < 0.005$. **e-g** Chem-036 blocked the aggregation of WT-SOD1 induced by TPEN. **e** Oligomerization of WT-SOD1 induced by TPEN was inhibited with Chem-036. After transfection with WT-SOD1 vectors in SK-N-SH cells, Chem-036 and TPEN was incubated for 12 h. The cells were harvested with TNN buffer and treated with 0.04% glutaraldehyde (Glu) for 1 h. SDS–PAGE was performed using the pan-SOD1 antibody. Actin was used as loading control. **f** Chem-036 reduced TPEN-induced WT-SOD1 inclusions. Cells were transfected with WT-SOD1 vectors for 24 h. After incubation, cells were treated with Chem-036 (5 μM) for 24 h and fixed with PFA. SOD1 expression was observed under fluorescence microscope with pan-SOD1 antibody (white arrows; strong intensity of SOD1). Scale bar; 10 μm. **g** Cells with SOD1 inclusions (white arrows; strong intensity of SOD1) were counted by photomicrographs and the percentages were shown with ± SD. $n = 3$ independent experiments; two-tailed Student's $t$-test, *$P < 0.05$, ***$P < 0.005$.

SOD1 [G93A-Tg] model mice, whereas no immunostaining is detected from WT mice (Fig. 4f, g). The SOD1 accumulation in model mice treated by PRG-A01 was reduced in the ventral region of the cervical spinal cord as well as in the lumbar spinal cord (Fig. 4f, g, and S7a–d). To check whether PRG-A01 treatment could contribute to the survival of spinal cord neurons, the spinal cord was sectioned and stained with hematoxylin and eosin (H&E). The number of the neurons were disappeared in the vehicle treated mice but were significantly maintained by PRG-A01 treated mice (Fig. S8a–d; black and yellow arrowheads, respectively). The abnormalities of myelin and demyelination which are related to the progression of ALS are increased in ALS patients[46]. Loss of myelin (yellow arrowheads) and nerve cells (black arrowheads) with Luxol fast blue (LFB) staining could be visible in vehicle-treated model mouse, but PRG-A01 treated model mice were recovered from the loss with comparable level to with wild type mice (Fig. S8e, f). In the previous report, neuronal markers (microtubule-associated protein 2; MAP2 and NeuN) are decreased in ALS patients and SOD1[G93A-Tg] model mouse[47–50]. Immunohistochemistry (IHC) analysis of MAP2 or NeuN showed that the marker positive cells were considerably increased in the PRG-A01 treated model mice compared to the vehicle treated model mice in both ventral and dorsal regions (Fig. 4h, i, and S9a–e). Finally, we investigated how PRG-A01 improved the survival rate of ALS model mouse. Although weight loss was not reversed by the treatment (Fig. 4b), PRG-A01 treated model mice showed fairly prolonged survival by a median of 10 to 150.3 days compared to a median survival of 140.5 days (7 % improvement) for vehicle treated one (Fig. 4j, S9f, g). Here, we suggest PRG-A01 has very impressive therapeutic effects by increasing muscle strength, motor neuron, mobility and life span.

## Discussion

ALS is one of the progressive neurodegenerative disorders with the accumulation of misfolding and aggregation protein such as AD and PD[1–3]. In this study, we supported the idea that the pathological accumulation of SOD1 can be one of the reasons for ALS disease. WT-SOD1 aggregation could be induced by MT-SOD1 through prion-like propagation, deregulation of TDP-43 and cellular stresses (Figs. 1, 3, S1, S3 and S4). We assume that ALS progression through TDP-43 mutation or overexpression might be related to SOD1 aggregation. Until now, several drugs for ALS therapy have been mentioned as candidates (Riluzole, Radicava), but they are not good effect on survival rate. Riluzole is known to be NMDA, GABA receptor inhibitor[51–53] and Radicava (Edaravone) shows the antioxidant effect[54,55], so these chemicals delay the ALS progression. However, it has been not a fundamental treatment for ALS therapy. So we tried to find new drugs for targeting aggregation or misfolding SOD1. According to a recent report, treatment of human–derived antibody that targets misfolded SOD1 delayed the onset of neurodegeneration, extended survival in SOD1 [G93A-Tg], SOD1 [G37R-Tg] mouse model[39]. Therefore, it means misfolded or aggregation of SOD1 is the substantial target for ALS therapy.

SOD1 is a major antioxidant that is important for preventing oxidative stress[36,37,42]. Maintaining the physiological dimer of SOD1 when treating the drugs would be an important part to minimize side effect through protecting antioxidant impairment. Therefore, we tried to search for MT-SOD1 specific chemicals to inhibit the interface between WT-SOD1 and MT-SOD1, which was formed through prion-like propagation property by MT-SOD1. We found a small molecule, PRG-A01, as the inhibitor of SOD1 aggregation and misfolding through ELISA-based chemical screening (Fig. 2 and S2). PRG-A01 showed the inhibitory effect for MT-SOD1 aggregation (Fig. 2b) and it significantly resolved the misfolding form of MT-SOD1. Especially, misfolded specific SOD1 Ab did not observe SOD1 in PRG-A01 treatment (Fig. 3). It has been known that the reduction of the disulfide bond and the removal of the metal ions are well known post-translational modification to decrease the dimeric stability[56–58]. However, the chemical structure of PRG-A01 suggests that PRG-A01 is not probable to react with the disulfide and to chelate the metal ion from SOD1 (Fig. S2e). It implies that PRG-A01 may have a feature to block or change the misfolded structure of SOD1, so keep its physiological structure, although a more detailed study would be needed on how PRG-A01 distinguishes between WT and MT-SOD1.

TDP-43 cytoplasmic inclusions in neurons and glia are the major pathological marker of ALS or Frontotemporal dementia (FTD)[14,15]. Cytoplasmic aggregation by TDP-43 overexpression could induce SOD1 aggregation (Fig. 3), although how TDP-43 is accumulated in the cytosol is unknown. TDP-43 overexpression could induce its translocation to cytoplasm, and the mislocalized TDP-43 might propagate prion-likely to WT-SOD1 (Fig. 3a, b). Remarkably, PRG-A01 blocked cytoplasmic accumulation of TDP-43 (Fig. 3a), and through native gel analysis, we could confirm that the aggregative association with SOD1 was broken by PRG-A01 (Fig. 3e). The more intensive study about the

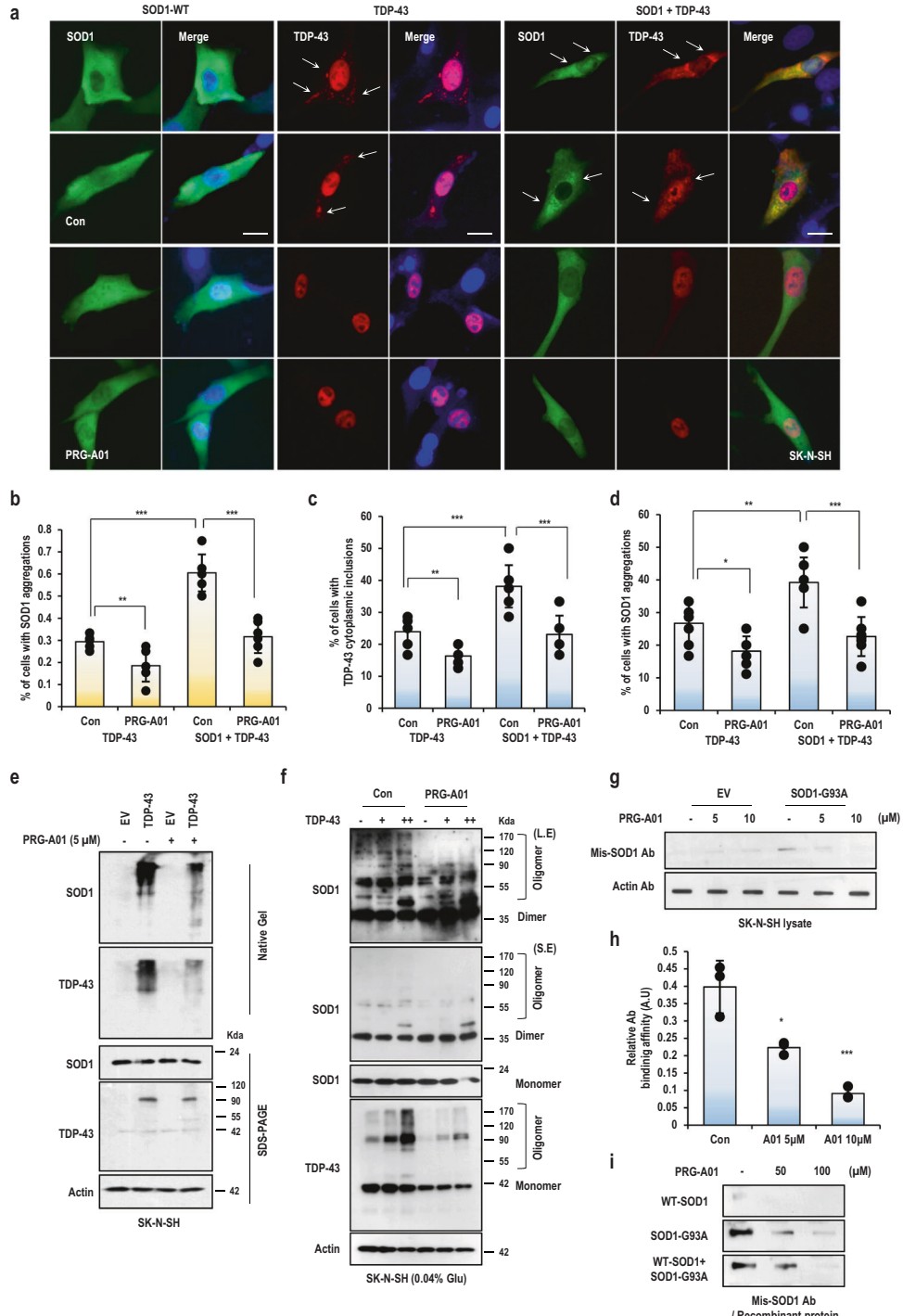

pathological propagation among ALS-related proteins, such as TDP-43, FUS and C9orf72, would be required to comprehend the cause of pathogenesis of sporadic ALS.

It has been known that the toxic aggregative or misfolding structures of SOD1 could induce the neuronal cell death[59]. In this study, PRG-A01 clearly inhibited human neuronal cell death induced by MT-SOD1 overexpression (Fig. S3c-f). Here, we suggest that PRG-A01 may provide the analytical basis for understanding an unknown structural interface of misfolded SOD1, which causes neuronal cell death through its aggregation.

Our chemical, PRG-A01, showed the protective effects on muscle weakness (49.1 % improvement, $p < 0.0121$) and movement disability (velocity 60.8% and distance 66.2% improvement,

$p < 0.0188$, $p < 0.00088$, respectively) shown in SOD1 $^{G93A-Tg}$ ALS model mouse (Fig. 4c–e and S5c–f), and histological analysis revealed that motor neuron viability in spinal cord was improved by 86 % ($p < 0.00015$) in PRG-A01 treated mice (Fig. S8a–d). The low expression of neuronal markers in SOD1 $^{G93A-Tg}$ ALS mouse was remarkably alleviated by PRG-A01 treatment compared to no treatment (MAP2: 125.1 % improvement, $p < 0.0011$ and NeuN: 86.4% improvement, $p < 0.035$) (Fig. 4h, i and S9a–e) and the pathological inclusions of SOD1 in neuronal cells was ameliorated by reduction of 35.5 % ($p < 0.014$) (Fig. 4f and S7). These promising results from PRG-A01 treated ALS model mice may explain that the improvement in both quantity and quality of mobility and the prolonged survival rate.

**Fig. 3 PRG-A01 blocks SOD1 aggregation and mis-folding. a** WT-SOD1 aggregation induced by ectopic TDP-43 expression was abolished by PRG-A01 treatment. Treatment of PRG-A01 could reduce SOD1 as well as TDP-43 aggregation. SK-N-SH cells were transfected with tdtomato tagged-TDP-43 and/ or WT-SOD1 for 24 h. After transfection, cells were incubated with PRG-A01 (5 μM) for 24 h. Cells were fixed with 4% PFA and stained with Pan-SOD1 antibody (Green) for monitoring SOD1 inclusion. Scale bar; 10 μm. **b** Cytoplasmic TDP-43 was diminished by PRG-A01 treatment. Cells with cytoplasmic TDP-43 (white arrows) among tdtomato positive ones were counted by photomicrographs and the cytosol/nucleus ratio were shown with ± SD. **c** TDP-43 inclusions induced its overexpression was blocked by PRG-A01 treatment. Cells with TDP-43 inclusions in cytoplasmic (white arrows; strong intensity of TDP-43) among tdtomato positive ones were counted and the percentages were shown with ± SD. **d** PRG-A01 reduced WT-SOD1 inclusion triggered by TDP-43 overexpression. Inclusion positive cells (white arrows; strong intensity of SOD1) were counted from photomicrographs and the percentages were shown with ± SD. n = 3 independent experiments; two-tailed Student's t-test, *P < 0.05, **P < 0.01, ***P < 0.005. **e** SOD1 high molecular complex with TDP-43 induced by TDP-43 ectopic expression were abolished by PRG-A01. Its treatment could reduce SOD1 as well as TDP-43 higher shifted band. SK-N-SH cells were transfected with TDP-43 and incubated with PRG-A01 (5 μM) for 24 h. After incubation, cells were harvested with TNN buffer and separated using native-gel analysis or SDS-PAGE. Actin was used for loading control. **f** PRG-A01 inhibited the oligomerized SOD1 and TDP-43 triggered by TDP-43 overexpression, without alternation of monomer-SOD1 expression. TDP-43 transfected cells were treated with PRG-A01 then harvested with TNN buffer and reacted with glutaraldehyde for detecting oligomerization of SOD1 and TDP-43. Actin was used for loading control. **g–h** PRG-A01 reduced misfolding formation of MT-SOD1. **g** For analysis of SOD1 conformation, cells transfected with G93A-SOD1 expressing vectors for 24 h and treated with dose dependent PRG-A01 for 24 h. After incubation, cells were lysed with TNN buffer and immobilized on nitrocellulose membrane using dot blot apparatus. The membrane was incubated with misfolding specific SOD1 or Actin antibody. Actin was used as the loading control. **h** Band density was quantified with Image J software and relative protein expression (mis-SOD1 Ab/Actin Ab) was calculated with ± SD. n = 3 independent experiments; two-tailed Student's t-test, A.U indicate arbitrary units. *P < 0.05, ***P < 0.005. **i** PRG-A01 suppressed misfolding of MT-SOD1 recombinant proteins. Recombinant WT (5 μg/ ml) and G93A-SOD1 (4 μg/ml) proteins were incubated with PRG-A01 (50, 100 μM) for 1 h in PBS, then transferred to a nitrocellulose membrane by dot blot apparatus and incubated with mis-folding specific SOD1 antibody.

Considering in vitro and in vivo results from our project, we suggest that this chemical, PRG-A01, is able to ameliorate the pathological progression of ALS disease and would be a useful candidate treatment for ALS therapy (Fig. 4k). Despite the therapeutic effect of PRG-A01, we showed the half-life of PRG-A01 in mouse blood was less than an hour (Fig. S9h). We presume that the linker domain of PRG-A01 would be easily broken after administration, then result in rapid degradation. Therefore, to overcome its unstable status, we have been generating about 20 derivatives through changing of linker domain or side chain of PRG-A01. Also, considering that the PRG-A01 did not result in weight gain, it is regarded as a limitation of i.p injection such as the number of injections and dosage. In order to maximize the effect of the drug, we are planning to feed the drug in food form daily starting at 6 weeks. We expect its derivatives will help ALS model mouse gaining the weight over the time points and prolonging the life span than PRG-A01, and it will be able to be examined whether the derivatives can delay the altered perivascular fibroblast activity in earlier than ALS disease onset as reported recently[60].

Neurodegenerative pathology is an important feature of ALS[25,26], which means that neural cell death is accumulated in the CNS through neural connection. A possible hypothesis is prion-like propagation in which mutant or misfolded SOD1 would be secreted to a neighbor neuron or microglia through the synapse[61,62]. Recently, many pathological evidences have been suggested that there were many neurodegenerative diseases with mixed proteinopathies[24]. Co-pathologies and cross-talks are likely a common feature with the nucleation process, which may involve oligomer formation accompanied by liquid−liquid phase separation among the oligomers of the (Aβ, tau), α-synuclein, and superoxide dismutase 1 proteins, which have been the mainstream concept underlying Alzheimer's disease, Parkinson's disease, and amyotrophic lateral sclerosis[16,24,27–32]. Since PRG-A01 was screened based on protein aggregation and misfolding of mutant SOD1, we might expect that it could resolve the pathologic interface formed by disordered regions in tau tangle and Aβ of AD and α-synuclein of PD[63–67].

Taken together, PRG-A01 efficiently inhibited or resolved the abnormal aggregation or misfolding induced by MT-SOD1, TDP-43 and cellular stresses. PRG-A01 ameliorates the pathological features of ALS model mouse, by improving the muscle strength, viability of motor neuron, mobility and survival. Our results suggest a PRG-A01 work as an effective candidate for ALS patients with SOD1 misfolding and aggregation.

## Materials and methods

**Transgenic mouse**. The experiments were performed in the Association for Assessment and Accreditation of Laboratory Animal Care certified facility, in compliance with animal policies approved by Pusan National University. B6SJL-Tg (SOD1 G93A) mice were obtained from Jackson Laboratory (Stock No: 002726). All mice were maintained under temperature and light-controlled conditions (20~23 °C, 12 h-12 h light/dark cycle) and provide autoclaved food and water.

**Drug treatment in vivo and histology analysis**. To evaluation the therapeutic effect of drug on life span, SOD1 G93A-Tg mice were administered with intraperitoneal injection of vehicle [DMSO, male (n = 5), female (n = 7)] or PRG-A01 [20 mg/kg, male (n = 6), female (n = 9)], twice per week from 12 weeks (days 80~85; disease onset) until end-stage phenotype. WT mice [male (n = 6), female (n = 5)] indicates negative littermates of the SOD1 G93A-Tg mice. Control mice were treated in the same conditions.

For histology analysis, SOD1 G93A-Tg male mice were treated with intraperitoneal injections of vehicle (n = 3) or PRG-A01 (n = 4) starting from 12 weeks and sacrificed at 18 weeks. WT male mice (n = 3) were sacrificed at the same time for positive control. After dissection of mice, the spinal cord was fixed using 4% paraformaldehyde for 48 h and embedded in paraffin blocks according to a basic tissue processing procedure. The embedded tissues (Cervical and Lumbar region of the spinal cord) were sectioned at 5 μm by Leica microtome and transferred onto adhesive-coated slides (Marienfeld laboratory glassware, Germany). After deparaffin and rehydration, slides were stained with hematoxylin and eosin (H&E) and Luxol fast blue (LFB) to detect the number of spinal cord neurons. For histology analysis of SOD1 G93A-Tg mouse pathologically, immunohistochemistry (IHC) with neuronal markers (MAP2, NeuN) and SOD1 was performed. The intensity of MAP2 and NeuN was calculated with Image J software. For counting intensity, IHC images were divided using the "color deconvolution" function in Image J software (National Institute of Health, NIH) and quantified the DAB signal. Counting was performed by three independent researchers.

**Motor performance assessment**. To evaluation the therapeutic effect of drug on grip strength, SOD1 G93A-Tg mice were treated with intraperitoneal injections of vehicle [male (n = 2), female (n = 2)] or PRG-A01 [20 mg/kg, male (n = 2), female (n = 3)] starting from 12 weeks (days 80~85; disease onset) and grip strength was checked every two weeks by using a grip strength meter. Age-matched WT mice [male (n = 2), female (n = 3)] were tested as positive control. The mouse was allowed to grip the tension bar with a fore-limb then pulled the tail slowly until its grip was broken. Each mouse was subjected to the test seven times to determine its forelimb strength. Each estimate given corresponded to average reads measured seven times, excluding the minimum and the maximum from each group of animals.

For the behavior test, SOD1 G93A-Tg mice were treated with intraperitoneal injections of vehicle [male (n = 2), female (n = 3)] or PRG-A01 [20 mg/kg, male

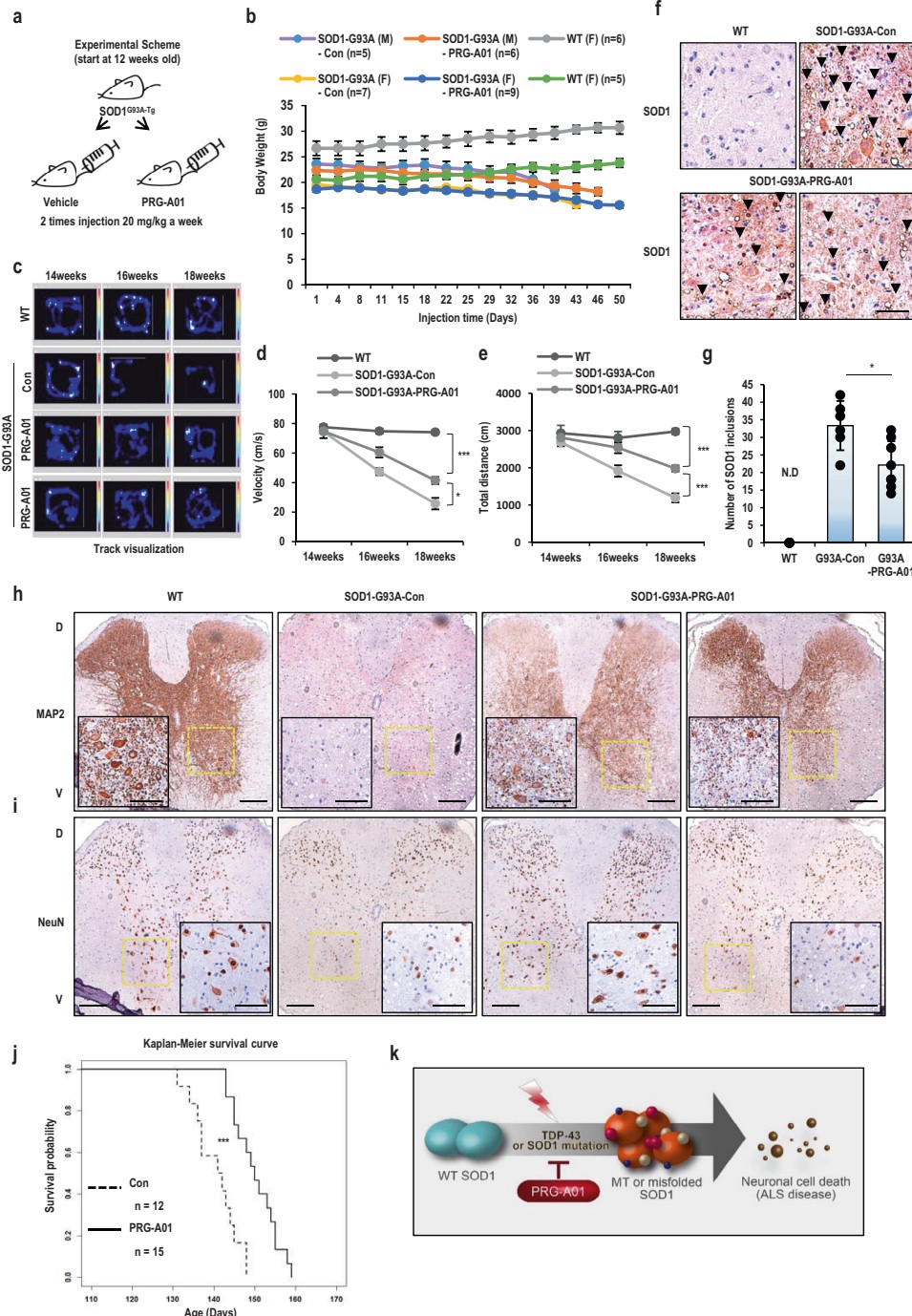

(n = 4), female (n = 6)] starting from 12 weeks. Using open field test, mouse movement was recorded every two weeks and recorded video analysis was performed at the endpoint of injection (18~20 weeks). Age-matched WT mice [male (n = 2), female (n = 3)] were tested as positive control. The recorded video file was analyzed for velocity, distance of movement using EthoVision XT 15 software (Noldus, USA).

**Cell culture and reagents**. HEK293 cells were obtained from the American Type Culture Collection (ATCC, Manassas, VA, USA) and maintained in DMEM media (10% fetal bovine serum and 1% penicillin-streptomycin) at 37 °C and 5% CO2. SK-N-SH and SK-N-MC cells were purchased from the Korean Cell Line Bank (KCLB, Seoul, South Korea). SK-N-SH cells were maintained in MEM medium (10% fetal bovine serum, 1% antibiotics, 25 mM HEPES and 300 mg/L L-Glu) and SK-N-MC cells were cultured with DMEM media (10% fetal bovine serum and 1% penicillin-streptomycin). Human fibroblast cell (9-year-old female) was obtained from the Coriell Cell Repositories (New Jersey, USA) and maintained in EMEM, containing 15% FBS, 2 mM Glutamine with 26 mM HEPES without antibiotics.

Thapsigargin (endoplasmic reticulum $Ca^{2+}$ depletor; CAS 67526-95-8) was purchased from Calbiochem (Darmstadt, Germany). $CoCl_2$ (hypoxia-inducer; C8661), $Zn^{2+}$ scavenger (TPEN; P4413) and BAPTA ($Ca^{2+}$ scavengers; A1076) were obtained from Sigma Aldrich (St, Louis, Mo, USA). $Cu^{2+}$ scavenger (ANT-224; CAS 649749-10-0) was purchased from Cayman Chemical (Michigan, USA)

**Chemical screening**. For chemical screening, we generated an ELISA assay system[40,41]. To select WT-SOD1 and mutant SOD1 binding inhibitor, we immobilized WT-SOD1 recombinant protein on a 96-well plate using 0.5% paraformaldehyde (PFA). After the plates were dried and washed with phosphate-buffered saline (PBS), we incubated with 50 μM of chemicals (final concentration), followed by adding mutant SOD1-GST (A4V, G37R, G85R, G93A) proteins. After 2 h reaction, the 96-well plates were washed with PBS and blocked by 3% skim milk to remove the background. The plates were incubated with anti-GST antibody (diluted in 1:10,000) for 1 h and then anti-mouse IgG-HRP (diluted in 1:50,000) for 1 h. After washing twice, plates were incubated with a 3,3′,5,5′-tetra-methylbenzidine (TMB) solution (Calbiochem) for 30 min and Stop solution (1 N

**Fig. 4 In vivo therapeutic effect of PRG-A01 on the SOD1$^{G93A\text{-}Tg}$ mouse model. a** Experimental scheme for chemical injection. 20 mg/kg of PRG-A01 i.p injection from 12 weeks until end-stage phenotype. **b** PRG-A01 did not show toxicity in body weight loss regardless of gender. **c** Open field test of SOD1$^{G93A\text{-}Tg}$ ALS model mice with PRG-A01 showed favorable effect on moving activity. SOD1$^{G93A\text{-}Tg}$ mice were injected con (DMSO, $n = 5$) or PRG-A01 ($n = 10$) and monitored mouse movement at 14, 16 and 18 weeks. Age-matched WT mice ($n = 5$) were tested as positive control. WT mice indicate negative littermates of SOD1$^{G93A\text{-}Tg}$ mice. Track visualization analysis was conducted by open field test and expressed as heatmap. Representative images were showed. **d–e** Velocity (**d**) and total distance (**e**) of mice carrying SOD1-G93A mutation was improved in PRG-A01 treated mice. Mice behavior which is shown in **c** was analyzed with the recorded video files. *$P < 0.05$, ***$P < 0.005$. **f–g** The inclusion of SOD1 were detected in the ventral horn of the cervical spinal cord from mice carrying SOD1-G93A mutation with IHC. **f** Comparing to DMSO-treated 18weeks old mice (Con, $n = 3$), reduction of SOD1 aggregation with SOD1-positive vacuoles (black arrowheads) were detected in PRG-A01 treated mice ($n = 4$). Representative images were showed and enlarged pictures of cervical and lumbar spinal cords were shown in Fig. S7a and c. 20x, Scale bar; 20 μm. The neuronal markers were maintained in PRG-A01 injected mice. **g** Number of SOD1 inclusion were counted with SOD1-positive vacuoles. N.D indicates not detectable. *$P < 0.05$. **h–i** The neuronal markers were maintained in dorsal and ventral horn of PRG-A01 injected mice ($n = 4$). The low expression of MAP2 (**h**) and NeuN (**i**) was detected in DMSO-treated control mice ($n = 3$). Representative images were showed with a magnification of 10x and an insert of 20x (yellow box). Enlarged pictures of cervical and lumbar spinal cords were shown in Fig. S9a and S9b. The intensity of neuronal markers was counted and plotted in Fig. S9c, d and e. Scale bar; 20 μm. **j** Kaplan–Meier survival curve of SOD1$^{G93A\text{-}Tg}$ ALS model mice. Comparing to DMSO-treated mice (Con, $n = 12$), PRG-A01-treated mice ($n = 15$) showed extended life span about 10 days. ***$P < 0.005$. **k** Diagram of misfolding SOD1 formation and working mode of PRG-A01.

H$_2$SO$_4$) for 30 min. Finally, we speculated the value by using the ELISA reader (absorbance at 450 nm). Negative control (-; red line) was incubated without mutant SOD1 and positive control (+; blue line) was incubated with mutant SOD1 without chemical.

**Recombinant proteins**. To produce the recombinant proteins, the human SOD1 (WT, A4V, G37R, G85R, G93A) were ligated into the *Eco*RI and HindIII sites of the pGEX-TEV vector, which is a modified vector made by adding a TEV protease cleavage site to pGEX-4T1 (Invitrogen). The recombinant proteins were expressed in the Escherichia coli (E.coli) strain BL21 (DE3) as GST-fusion proteins. The proteins were purified by glutathione-affinity chromatography.

**Western blot analysis**. Proteins were extracted from cells with RIPA buffer (50 mM Tris-Cl, pH 7.5, 150 mM NaCl, 1% NP-40, 0.1% SDS and 10% sodium deoxycholate) for SDS-PAGE and lysis buffer (50 mM Tris-Cl, pH 7.5, 150 mM NaCl, 0.3% NP-40; TNN buffer) for Native-PAGE. Samples were separated via SDS-PAGE or Native-PAGE and transferred to PVDF membrane. Blotted membranes were blocked by 3% skim milk containing TBST buffer for 1 h and incubated with specific antibodies. Reacted antibody was detected with ECL and X-ray film exposure. The following antibodies were used in this study; pan-SOD1 (1:2000, GTX100554) was purchased from Genetex (California, USA). Misfolded SOD1 specific antibody (1:2000, B8H10) was obtained from MediMabs (Montreal, Canada). Actin (1:2000, sc-1616), GST (1:3000, sc-138), GFP (1:2000, Green fluorescent protein; sc-8036) were purchased from Santa Cruz biotechnology (Santa Cruz, CA, USA). TDP-43 antibody (1:4000, 10782-2-AP) was obtained from Proteintech (Rosemont, IL, USA). Anti-FLAG (1:2000, Sigma; F3165) was provided by Sigma Aldrich (St, Louis, Mo, USA), HRP-conjugated goat anti-mouse, goat anti-rabbit and mouse anti-goat antibodies (Pierce, Thermo Fisher Scientific, Inc., Rockford, IL, USA) were used as secondary antibodies.

To isolation insoluble form of SOD1, SK-N-SH and HEK293 cells were transfected with WT and/or MT-SOD1 vectors. After 72 h incubation, the cells were treated with or without chemicals then its harvested with TNN buffer (50 mM Tris-Cl, pH 7.5, 150 mM NaCl, 0.3% NP-40) and centrifuged at 14000 rpm for 30 min then pellet (insoluble) and supernatant (soluble) were separated. Input indicated whole cell lysates (WCL) which were harvested with RIPA buffer.

To detect oligomeric status of SOD1, proteins were extracted from transfected cells with lysis buffer (50 mM Tris-Cl, pH 7.5, 150 mM NaCl, 0.3% NP-40, 100 mM iodoacetamide; TNNI buffer) and reacted with 0.04% glutaraldehyde for 1 h. The samples were performed with SDS-PAGE. Monomer and oligomer indicated SOD1 formation.

**Dot blot analysis**. To detect misfolded SOD1 expression, transfected cells with SOD1 vectors were treated with chemicals for 24 h. After incubation, cells were lysed with TNN buffer (50 mM Tris-Cl, pH 7.5, 150 mM NaCl, 0.3% NP-40) not containing detergent then cell lysates were immobilized on nitrocellulose membrane using the Bio-Dot SF Microfiltration apparatus (Bio-Rad Laboratories, Hercules, CA). In the case of peptide reaction, SOD1 recombinant proteins were reacted with the chemical for 1 h then samples were loaded on the membrane. Each membrane was washed with TBS and blocked by 3% skim milk to remove background for 1 h. After blocking, the membrane was incubated with misfolded SOD1 (1:2000) or Actin antibody (1:10000 in 1% skim milk containing TBST) for 30 min, then reacted with secondary antibody (goat anti-mouse IgG-horseradish peroxidase, 1:50,000 in 1% skim milk blocking buffer) for 30 min. Reacted antibody to proteins was detected with ECL and X-ray film exposure. Actin was used as the loading control.

**Immunofluorescence staining**. Cells on coverslips were washed with PBS and fixed with 4% PFA for 30 min at room temperature and then permeabilized in 0.1% Triton X-100/PBS for 10 min. After cells were treated with blocking solution (anti-Human Antibody diluted 1:500 in PBS) for 1 h, cells were incubated with anti-pan SOD1 (diluted in 1:400) for overnight at 4 °C. Finally, the cells were incubated with FITC-conjugated secondary antibodies at 4 °C for 6 h. The nucleus was stained with 4, 6-diamidino-2-phenylindole (DAPI) and the endoplasmic reticulum (ER) was stained using ER-Tracker Red dye for 10 min. Cells were washed three times with PBS, then the coverslips were mounted with the mounting solution (H-5501; Vector Laboratories (Burlingame, CA, USA)) and analyzed by fluorescence microscopy (Zeiss).

**Transfection of vectors**. GFP-SOD1 (WT; #26402, G85R; #26405, G93A; #26406), non-tagged SOD1 (WT; #26397, A4V; #26398, G37R; #26399, G85R; #26400, G93A; #26401), tdTomato-TDP43 expression vectors (#28205) were purchased from Addgene (Cambridge, MA, USA). Transfection was performed using Jet-PEI reagent or Jet-Optimus (JetPEI and JetOptimus; Polyplus transfection, New York, NY, USA) according to the manufacturer's protocol. Briefly, the vector was mixed with JetPEI reagent in 150 mM NaCl buffer or JetOptimus reagent in JetOptimus buffer, then the mixture was incubated for 15 min. The mixture was added to cells in serum-free medium for 4 h. After incubation, cells were replaced with a culture medium supplemented with 10% FBS

**Measurement of cell viability**. To examine the cell viability, cells were incubated with 0.5 mg/ml of MTT solution (475989; Merck, Darmstadt, Germany) for 4 h at 37 °C. After removing excess solution and washing with PBS, the precipitated materials were dissolved in 200 μl DMSO and quantified by measuring absorbance at 540 nm. To evaluate the cell death, cells were transfected with SOD1 expressing vectors (WT and MT) in SK-N-MC or SK-N-SH cells for 48 h. After transfection, cells were treated with chemicals (chem-036) for 48 h. We performed propidium iodide (PI) staining for detecting dead cells without fixation for 30 min. Then, cells were fixed with 4% PFA and stained with DAPI for monitoring total cells. PI intensity was calculated through the "color histogram" function of the Image J software (National Institute of Health, NIH) with randomly selected fields in images.

**Statistics and reproducibility**. The student's *t*-test was used for comparisons of two groups. *P*-value less than 0.05 was considered significant. Error bars indicate standard deviation (SD). Data for all figures are expressed as means ± SD of at least three independent experiments. Detail of statistical analysis and number of replicates (*n*) can be found in the figure legends.

**Reporting summary**. Further information on research design is available in the Nature Research Reporting Summary linked to this article.

## Data availability
Uncropped blots are shown in Supplementary Information. Source data underlying plots shown in figures are provided in Supplementary Data 1. Mouse behavior datas are shown in Supplementary Movie. All other data that support the findings of this study are available from the corresponding author upon reasonable request.

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

## Acknowledgements

This work was supported by National Research Foundation of Korea (NRF) through a grant funded by the Korea government (MSIT) (NRF-2020R1A4A1019322 to B.J.P., NRF-2019R1C1C1007803). The research was supported by PRG S&Tech Inc.

## Author contributions

T.G.W., M.H.Y., S.M.K., S.Y.P., J.H.C., and Y.J.H. performed the experiments. T.G.W., E.M.J., N.C.H., B.H.K. and B.J.P. conceived the experimental designs and interpretation of data and revised the manuscript. J.S.A. and N.C.H. contributed to recombinant proteins synthesis. H.W.J., Y.J.S. and Y.H.K. synthesized and offered the chemicals. T.G.W., M.H.Y., Y.J.H. and E.M.J. contributed to the in vivo experiments and analysis.

## Competing interests

The authors declare no competing interests.
