## [Transparent Peer Review File · Communications Biology]

Reviewers' comments:

Reviewer #1 (Remarks to the Author):

The authors present a novel chemical inhibitor against SOD1 misfolding and aggregation that protects neuron-loss and ameliorates disease symptoms in ALS model.

This is a very nice study and the manuscript is well-written. I only have four minor questions to improve the quality of the manuscript.

1. The authors used a chemical screening system based on ELISA with their proven system and chemical library. They identified chem-036 renamed as PRG-A01. What is the chemical structure of this molecule?
2. The crystal structures of native SOD1 dimer is known. It would be interesting that the authors use state-of-the art protein-inhibitor docking WEB-servers to identify probable binding sites.
3. It is well known that post-translational modifications affect the dimer stability. Is PRG-A01 going to be effective with these SOD1 modifications? This aspect should be discussed.
4. In the conclusion, the authors wrote 'we have been generating PRG-A01-related derivatives'. How they differ from PRG-A01? Fig S10 is a very important result and should be shown in the main text unless only four figures are allowed.

Reviewer #2 (Remarks to the Author):

This is very interesting article that explore the therapeutic effect of PRG-A01 for ALS patients with SOD1 misfolding and aggregation. The authors use both human neuroblastoma cell line expressing mutant SOD1 and mtSOD1 fALS mouse models to validate their finding. The authors showed WT-SOD1 aggregation could be induced by MT-SOD1 through prion-like propagation, deregulation of TDP-43 and various cellular stresses. They suggest that PRG-A01 may provide the analytical basis for understanding an unknown structural interface of misfolded SOD1, which causes neuronal cell death through its aggregation. The presented work requires further improvement before consideration of publication. Several issues need to be addressed as follows:

1. Page 2 Main part authors say: "About 10-20% of patients show inherited patterns (familial ALS; fALS), whereas the remaining ~." What is the reference for 10-20%? It should be 5-10%.
2. In figure 1, authors showed that the overexpression of non-tagged-WT SOD1 turned soluble GFP-WT-SOD1 into insoluble aggregation. Why is the GFP-WT SOD1 aggregation also formed by WT SOD1? On the other hand, in figure 2b, please explain why figure doesn't show insoluble aggregation (oligomer formation) in WT SOD1.
3. In western blotting sample preparation, please explain in detail the insoluble sample preparation process in MATERIALS AND METHODS section.
4. How are SK-N-SH and SK-N-MC (figure S3e) cells different?
5. How many experiments in each sample were prepared? All figure panels where there is quantification should specify the "n" and the statistical analyses used.
6. In figure 4, authors should specify whether wild-type (WT) SOD1 mice refers to transgenic mice expression human SOD1 transgene or are in fact negative littermates of the G93A-SOD1 mice.
7. The authors said that SOD1 G93A-Tg mice were administered with con (DMSO), PRG-A01 (20 mg/kg) by intraperitoneal injection, twice per week from 12 weeks old mice for 6~8 weeks. I wonder if you started injection from 12 week old mice in animal experiments. In animal model of ALS, the timing of drug injection is thought to be rather late.

8. What are the numbers of animals use for each of the studies? What are the numbers of male and female animals use for each of the studies? Differences between transgenic male and female mice have been described in several studies, not only about weight or survival, but also about onset and progression disease. Please write in detail in MATERIALS AND METHODS section.

Dear Editor of Communication Biology

Here we provided our response for reviewer's critics as point-by-point form. Bold letters are our response.

Thanks again for evaluation of our manuscript.

Point-by-Point response for reviewer's critics

Reviewer #1 (Remarks to the Author):

The authors present a novel chemical inhibitor against SOD1 misfolding and aggregation that protects neuron-loss and ameliorates disease symptoms in ALS model.

This is a very nice study and the manuscript is well-written. I only have four minor questions to improve the quality of the manuscript.

➤ **Thank you for your positive answer**

1. The authors used a chemical screening system based on ELISA with their proven system and chemical library. They identified chem-036 renamed as PRG-A01. What is the chemical structure of this molecule?

➤ **We added the chemical structure of PRG-A01 (Chem-036) to Supple. Fig 2d**

➤ **Supple. Fig 2d and 2e was newly arranged**

2. The crystal structures of native SOD1 dimer is known. It would be interesting that the authors use state-of-the art protein-inhibitor docking WEB-servers to identify probable binding sites.

➤ **As you suggested, we tried to identify the probable binding sites using the program PyRx. The inhibitor was bound to the different sites on the surface of the dimeric structure of the native SOD1.**

➤ **We guess that the inhibitor selectively binds to the unknown interface of aggregated/misfolded SOD1**

➤ **We added the figure about probable binding site of PRG-A01 at Supple. Fig 2e**

➤ **We addressed it in the result and discussion section (Page 5; line 25-27, Page 6; line 1-3)**

3. It is well known that post-translational modifications affect the dimer stability. Is PRG-A01 going to be effective with these SOD1 modifications? This aspect should be discussed.

➤ **The reduction of the disulfide bond and the removal of the metal ions are well known post-translational modification to decrease the dimeric stability (Chattopadhyay M. et al. J Biol Chem. 2015, Oztug Duzer ZA. Et al. Plos One. 2009, Banci L. et al, PNAS. 2007).**

- **However, as shown in Supple Fig 2e, the chemical structure of PRG-A01 suggests that PRG-A01 is not probable to react with the disulfide and to chelate the metal ion from SOD1.**
- **We described it in the discussion section (Page 10, line 18-21)**

4. In the conclusion, the authors wrote 'we have been generating PRG-A01-related derivatives'. How they differ from PRG-A01? Fig S10 is a very important result and should be shown in the main text unless only four figures are allowed.

- **It seems that PRG-A01 showed unstable status due to linker domain**
- **We generated derivatives about 20 candidates through changes the linker domain or side chain of PRG-A01. As a result, we selected the 3 lead compound which showed the improved pharmacokinetics.**
- **Considering our result, we anticipate that PRG-A01 derivatives show the improved therapeutic effect on prolong the life span, alleviation in disease symptoms.**
- **We described it in the discussion section and are now preparing separated manuscript.**
- **As you suggested, we rearranged the Fig S10, body weight data of male and female (Supple. Fig 10a and 10b) at Main Fig 4b and described it in results section and figure legend. (Page 8; line 12)**
- **Also, we arranged the pharmacokinetics data (Fig S10c) at Supple. Fig 9h.**

Reviewer #2 (Remarks to the Author):

This is very interesting article that explore the therapeutic effect of PRG-A01 for ALS patients with SOD1 misfolding and aggregation. The authors use both human neuroblastoma cell line expressing mutant SOD1 and mtSOD1 fALS mouse models to validate their finding. The authors showed WT-SOD1 aggregation could be induced by MT-SOD1 through prion-like propagation, deregulation of TDP-43 and various cellular stresses. They suggest that PRG-A01 may provide the analytical basis for understanding an unknown structural interface of misfolded SOD1, which causes neuronal cell death through its aggregation. The presented work requires further improvement before consideration of publication. Several issues need to be addressed as follows:

- **Thank you for taking your time to review our paper.**

1. Page 2 Main part authors say: "About 10-20% of patients show inherited patterns (familial ALS; fALS), whereas the remaining ~." What is the reference for 10-20%? It should be 5-10%.

- **As you suggested, we corrected the sentence (Page 2; line 20)**

2. In figure 1, authors showed that the overexpression of non-tagged-WT SOD1 turned soluble GFP-WT-SOD1 into insoluble aggregation. Why is the GFP-WT SOD1 aggregation also formed by WT SOD1? On the other hand, in figure 2b, please explain why figure doesn't show insoluble aggregation (oligomer formation) in WT SOD1.

- **It has been known that inclusion of WT-SOD1 are detected in sporadic ALS patients and mouse models with WT-SOD1 overexpression. (Graffmo, K.S. et al. Human Molecular Genetics. 2013, Furukawa Y. et al. Translational neurodegeneration. 2020, Jaarsma D. et al, Neurobiology of Disease. 2000)**
- **We described it in the introduction sections and added the references (Page 3; line 1-2)**
- **We confirmed that WT-SOD1 overexpression induce WT-SOD1 aggregation (Fig1a, 1c and Supple. Fig 2a-2c)**
- **However, as shown in fig 1a,1c and Supple. fig 2b, WT-SOD1 aggregation formed by WT-SOD1 overexpression is weaker than induced by MT-SOD1**
- **In addition, based on cell death assay, as shown in Supple. Fig S3c-3f, WT-SOD1 overexpressing cells did not show cell death than MT-SOD1 overexpressing cells.**
- **So, we guess that inclusion of WT-SOD1 by WT-SOD1 overexpression, would be resolved easily than MT-SOD1**
- **As a result, WT-SOD1 aggregation was resolved during SDS-PAGE, so oligomer formation was not barely detected**
- **However, In Furukawa Y. et al, (Translational neurodegeneration. 2020) they showed that post-translational modification converts WT-SOD1 to misfolding or aggregated SOD1**
- **As shown in fig 1e-1g, Supple. Fig 1e-g, we confirmed that cellular stress induced WT-SOD1 aggregation**
- **Moreover, as shown Fig 2e, WT-SOD1 oligomer was detected by treated with Zn ion chelator (TPEN).**
- **We guess that stress-induced WT-SOD1 aggregation is more solid form through changing the structure than WT-SOD1 overexpression**
- **Although further studies are needed, understanding the WT-SOD1 aggregation will be a important explanation of the pathogenesis of sporadic ALS**

3. In western blotting sample preparation, please explain in detail the insoluble sample preparation process in MATERIALS AND METHODS section.

- **We added the detailed explanation for sample preparation (Page 16, line 17-25)**

4. How are SK-N-SH and SK-N-MC (figure S3e) cells different?

- **SK-N-SH and SK-N-MC are human neuroblastoma cell lines. SK-N-SH cells have dopamine-beta-hydroxylase activity compared to SK-N-MC**
- **Both cells are used for studying neuronal disease model (for example, ALS, Parkinson's disease, Alzheimer's disease)**

- **We added the figure about cell death (PI staining) induced by SOD1-overexpression in SK-N-SH cells (Supple. Fig 3c-3d)**

5. How many experiments in each sample were prepared? All figure panels where there is quantification should specify the "n" and the statistical analyses used.

- **We do experiments more than 3 times independently, repeatedly**
- **We described it in the figure legend**
- **We added the graphs with quantification and statistical analysis of IF experiments in the section of Supplementary figures (for example Supple. Fig 1a, 1c, 1d and 1f)**

6. In figure 4, authors should specify whether wild-type (WT) SOD1 mice refers to transgenic mice expression human SOD1 transgene or are in fact negative littermates of the G93A-SOD1 mice.

- **WT-Mice indicated the negative littermates of the G93A-SOD1 mice**
- **We added the detailed information about mice at figure legend and material and method sections (Page 13; line 10-17)**

7. The authors said that SOD1 G93A-Tg mice were administered with con (DMSO), PRG-A01 (20 mg/kg) by intraperitoneal injection, twice per week from 12 weeks old mice for 6~8 weeks. I wonder if you started injection from 12 week old mice in animal experiments. In animal model of ALS, the timing of drug injection is thought to be rather late.

- **It has been known that the therapeutic effect of drug started from disease onset (11~13 weeks) using SOD1-G93A tg mouse model. (Vallarola A. et al. Journal of Neuroinflammation. 2018, Ly D. et al. Purinergic Signal. 2020, Hogg M,C. et al. Amyotroph Lateral Scler Frontotemporal Degener. 2018, Riluzole)**
- **However, as you reviewed, we agree that the disease is progressing in 12~13 weeks old G93A-SOD1 mice, and as a result, it is rather late to monitor the therapeutic effect of drug**
- **We tried i.p injection starting from 8 weeks, but they were too young and small to tolerate the injection stress**
- **Although, we started the i.p injection twice per week from 11~12 weeks (disease onset) for reducing injection stress, its showed therapeutic effect**
- **i.p injection has a limitation for monitoring effect of drug, such as the number of injections and dosage.**
- **Next, in order to overcome its limitation and maximize the effect of the drug, we are considering making the drugs in the form of food and trying to feed them daily starting at 6 weeks**
- **We described it in the discussion section (Page 11; line 25-26, Page 12; line 1-5)**

8. What are the numbers of animals use for each of the studies? What are the numbers of male and

female animals use for each of the studies? Differences between transgenic male and female mice have been described in several studies, not only about weight or survival, but also about onset and progression disease. Please write in detail in MATERIALS AND METHODS section.

- **We added the lifespan according to gender at Supple. Fig 9f and 9g**
- **SOD1-G93A-tg male mouse showed an early onset and progression about 5~7 days compared to female mouse.**
- **In addition, vehicle-treated female mouse (142.9 days) had a lifespan about 5 days longer than that of male mice (137.2 days).**
- **PRG-A01-treated mice showed extended lifespan about 13 days (Male, 137.2 -> 150.5 days) and 8 days (Female, 142.9 -> 150.1 days) respectively**
- **We described the detailed information about numbers and genders of mouse for each experiment in material and methods section and figure legends. (Page 13; line 10-17, Page 14; line 5-17)**

REVIEWERS' COMMENTS:

Reviewer #1 (Remarks to the Author):

to be published. All my comments have been addressed.

Reviewer #2 (Remarks to the Author):

You have taken my comments with great consideration in the revised version of the paper. The manuscript is suitable for publication.